# Observation of electronic modes in open cavity resonator

Hwanchul Jung[1,2,8], Dongsung T. Park [3,8], Seokyeong Lee [3], Uhjin Kim[4], Chanuk Yang[4], Jehyun Kim[5], V. Umansky[6], Dohun Kim [5], H.-S. Sim[3], Yunchul Chung [1,2,9] ✉, Hyoungsoon Choi [3,7,9] ✉ & Hyung Kook Choi [4,9] ✉

The resemblance between electrons and optical waves has strongly driven the advancement of mesoscopic physics, evidenced by the widespread use of terms such as fermion or electron optics. However, electron waves have yet to be understood in open cavity structures which have provided contemporary optics with rich insight towards non-Hermitian systems and complex interactions between resonance modes. Here, we report the realization of an open cavity resonator in a two-dimensional electronic system. We studied the resonant electron modes within the cavity and resolved the signatures of longitudinal and transverse quantization, showing that the modes are robust despite the cavity being highly coupled to the open background continuum. The transverse modes were investigated by applying a controlled deformation to the cavity, and their spatial distributions were further analyzed using magnetoconductance measurements and numerical simulation. These results lay the groundwork to exploring matter waves in the context of modern optical frameworks.

The profound resemblance between optical waves and ballistic electrons has inspired numerous electronic realizations of optical elements such as lenses[1–3], beam splitters[4–7], and their implementation towards various electron interferometers[8–17]. In particular, the interference between electronic quasiparticles has become an especially important concept when probing various exotic properties of many-body excitations, e.g., anyon braiding in fractional quantum Hall excitations[18–23]. Such interferences arise naturally in optical physics under the name of cavity resonances[24–26], and recent developments in optics have emphasized the role of cavity modes in understanding integrable ray dynamics and waves in open systems[27–30]. Although mode resonances in cavity-like structures have also been studied in electronic systems, much research had only focused on nearly-closed

structures, e.g., quantum dots (QDs) and quantum billiards[31–33]. Even in the celebrated quantum Hall Fabry–Pérot configuration[12,18], the edge states serve a purpose similar to fiber optic cables where transverse motion is strongly restricted. However, the richness of mode dynamics in optical cavities arises from the availability of multiple resonator dimensions and a deliberate openness of the cavities[28–30,34]. While such optical cavity modes are well-understood, their adaptation to matter waves such as electrons is a nontrivial problem that has yet to be investigated.

Here, we report the realization of an open cavity resonator in a GaAs/AlGaAs two-dimensional electron gas (2DEG) using curved split-gates to define the cavity mirrors. Electrons were injected into the cavity using tunnel-coupled quantum point contacts (QPCs) at the

[1]Department of Physics, Pusan National University, Busan 46241, Republic of Korea. [2]Quantum Matter Core-Facility, Department of Physics, Pusan National University, Busan 46241, Republic of Korea. [3]Department of Physics, KAIST, Daejeon 34141, Republic of Korea. [4]Department of Physics, Research Institute of Physics and Chemistry, Jeonbuk National University, Jeonju 54896, Republic of Korea. [5]Department of Physics and Astronomy, and Institute of Applied Physics, Seoul National University, Seoul 08826, Republic of Korea. [6]Department of Condensed Matter Physics, Weizmann Institute of Science, Rehovot 76100, Israel. [7]KI for the Nano Century, KAIST, Daejeon 34141, Republic of Korea. [8]These authors contributed equally: Hwanchul Jung, Dongsung T. Park. [9]These authors jointly supervised this work: Yunchul Chung, Hyoungsoon Choi, Hyung Kook Choi. ✉e-mail: ycchung@pusan.ac.kr; h.choi@kaist.ac.kr; hkchoi@jbnu.ac.kr

center of the mirrors, and a modulation gate covering the cavity region was used to control the electron wavelength. Robust cavity resonances were observed from conductance measurements despite the strong coupling of the cavity to its open sides. We show that the main periodicity of the resonances can be attributed to the longitudinal Fabry–Pérot modes by analyzing the conductance lineshape and resonance energy spectrum. Furthermore, we demonstrated the tunability of transverse modes by introducing a cavity deformation via the mirror split-gates. The cavity deformation induced a detuning among the Fabry–Pérot resonances, and the spatial distribution of the transverse modes were measured through the cavity magnetoconductance. With the aid of tight-binding simulations, we identified that the observed transverse modes came in two variants: one lying on top of the cavity axis, and another lopsided to one side of the axis. Having established the modal nature of the cavity resonances, we discuss the distinguishing characteristics of the demonstrated system and its future applications.

## Results

### Electronic cavity resonator

An optical cavity resonator often refers to the Fabry–Pérot resonator, where longitudinal cavity modes form due to constructive interferences along the cavity axis[25,26]. Although the archetypical Fabry–Pérot interferometer is constructed using plane-parallel mirrors (Fig. 1a), such an arrangement suffers from an instability where small misalignments can easily allow the optical rays to escape through the cavity sides, spoiling the cavity resonance. Instead, the flat mirrors can be replaced with mirror lenses that refocus the diverging rays back towards the cavity axis (Fig. 1b), raising the likelihood that the rays stay within the cavity. For mirror radii of $R_1$ and $R_2$, ray optics predicts that the cavity is stable if $0 \leq (1 - L/R_1)(1 - L/R_2) \leq 1$ where $L$ is the cavity length. The lensing action of the cavity mirrors provides an effective potential in the transverse direction[35], and the resulting confinement, despite the absence of explicit boundaries, is a defining quality of the cavity resonator. The resonant modes of the open cavity are thus described by standing waves in two directions: the transverse modes, also known as Hermite–Gaussian modes, and longitudinal modes, which are usually referred to as Fabry–Pérot interferences. In a cavity resonator, multiple Fabry–Pérot spectra may be seen simultaneously when multiple transverse modes are occupied[34,36], each with their family of longitudinal modes.

The cavity resonator was implemented using electron waves in a mesoscopic platform, as illustrated in Fig. 1c. The cavity was defined and manipulated by Schottky gates on top of the GaAs/AlGaAs heterostructure with a 2DEG. Figure 1d is a false-colored scanning electron microscope image of the device with the gates colored in yellow. The mirror gates 1u-1d and 2u-2d defined the cavity geometry, and the modulation gate M controlled the electron wavelength within the cavity region. Specifically, the mirror gates were biased with a negative voltage in order to locally deplete the 2DEG directly underneath the gates, shown as the white area in Fig. 1e, acting as reflectors to the electron waves. The undepleted region between the 1u-1d (2u-2d) mirror gates formed a quantum point contact[37–39] (QPC) with conductance $g_1$ ($g_2$), normalized in units of $G_0 = 2e^2/h$, which is equivalent to the optical transmission coefficient. The mirror QPCs were tuned using their gate voltages and functioned as partially transmitting sections of the mirrors when operated in the tunneling regime, i.e., $g_{1,2} < 1$. During operation, electrons injected through reservoir S may either return back to the source, transmit to the drain D, or escape towards the open cavity sides O1 and O2. The transmission spectra were obtained by measuring the conductances between reservoirs while modulating the wavelength of the cavity electrons by scanning through gate voltage $V_M$, which linearly modulates the Fermi energy[40,41].

Experiments were performed in the conventional mesoscopic transport setup. The 2DEG sample with carrier density $n = 2.3 \times 10^{11}$ cm$^{-2}$ was bias cooled[42] in a homemade dilution refrigerator with a base temperature < 150 mK. The cavity was defined with dimensions $R_{1,2} = 350$ nm and $L = 500$ nm, satisfying the aforementioned stability condition. The differential conductances $G$ of the devices were measured using a homemade preamplifier[43] and a lock-in amplifier with an excitation source of 10 μV$_{rms}$ at 489 Hz. All conductances presented here are given in their normalized forms $g = G/G_0$ in order to emphasize their role as transmission coefficients[44,45].

### Open cavity modes

Figure 2 shows the principal transmission properties of the cavity device. The cavity transmission $g_{cav}$ and loss $g_{loss}$ were measured in

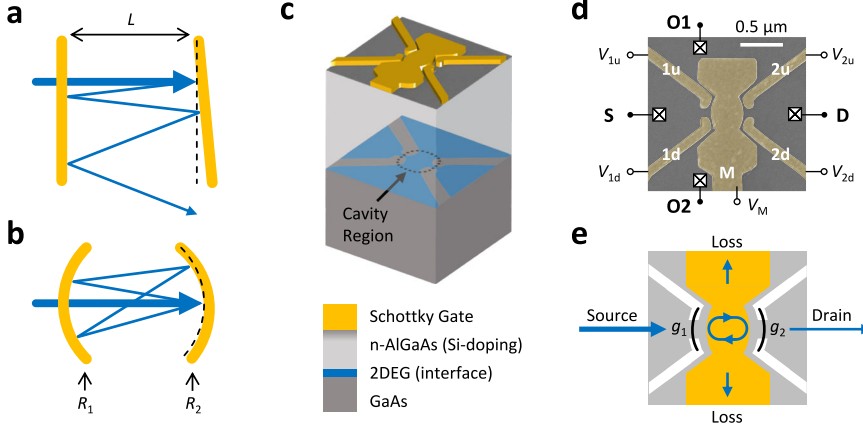

**Fig. 1 | Electronic cavity resonator. a** The archetypical Fabry–Pérot Interferometer, a plane-parallel resonator, is unstable as a small misalignment allows the rays to leak through the sides. **b** A stable cavity resonator constructed using concave mirrors provides effective transverse confinement through the mirror curvatures. **c** Electronic cavity resonator fabricated on a GaAs/AlGaAs heterostructure. Electrons from the Si-doping layer in the n-AlGaAs migrate to the semiconductor interface and form a two-dimensional electron gas (2DEG, blue), where the electrostatic potential is shaped by the Schottky gates on top (yellow). **d** False-colored image of the Schottky gates on a typical device. Gates 1u-1d (2u-2d) define the left (right) mirror of the cavity, and gate M modulates the wavelength of electrons within the cavity. The 2DEG is accessed through the ohmic contacts attached to the four reservoirs S, D, O1, and O2. **e** Electron mirrors (black curve) are formed by negatively biasing the gates, which locally deplete the 2DEG (white area), and tunneling contacts with normalized conductances $g_{1,2} < 1$ at the center of each mirror function as partially transmitting portions of the mirror. Electron wavelength in the cavity is modulated by voltage $V_M$ which locally tunes the Fermi energy (yellow area).

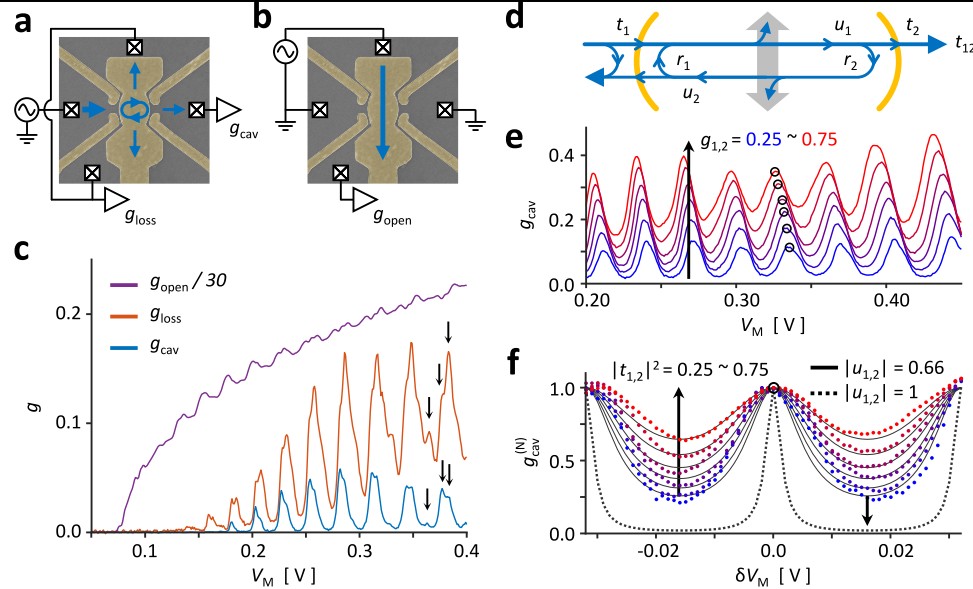

**Fig. 2 | Open cavity modes. a** Transmission properties of the cavity resonator were measured by the conductances $g_{cav}$ and $g_{loss}$. **b** Openness of the cavity region was given by $g_{open}$ which measures the transmission in the direction transverse to the cavity (blue arrow). **c** Cavity transmissions $g_{cav}$ and $g_{loss}$ measured against $V_M$ shows coinciding resonance peaks, implying the presence of circulating waves, i.e., cavity modes. The longitudinal nature of the modes can be seen by the quasiperiodicity of the conductance peaks, and hints of the transverse modes can be seen by the fine peak structures (arrows). These characteristics were observed while the cavity stayed highly open to its sides, as seen from $g_{open}$. **d** The mesoscopic cavity

resonator model was constructed using the Landauer–Büttiker formalism. Cavity transmission $g_{cav}$ is given by the total transmission across both mirrors $|t_{12}|^2$, and the openness is accounted by subunitary propagation across the mirrors ($|u_i|<1$). **e** Changes in the mirror conductance $g_i = 0.25 \sim 0.75$ led to uniform changes across the cavity spectra. A closer inspection of the circled peaks is shown in **f** after normalization. The model prediction for $|t_i|^2 = 0.25 \sim 0.75$ and $|u_i| = 0.66$ has been plotted as solid black lines, and the prediction for $|t_i|^2 = 0.25$ in a closed cavity ($|u_i| = 1$) has been plotted as a dotted black line for comparison.

configuration Fig. 2a, and the openness in the transverse direction was gauged by measuring the conductance $g_{open}$ between the cavity sides as shown in Fig. 2b. Since the gate M directly affects the mirror conductances $g_{1,2}$, the QPC gate voltages were adjusted to compensate for unwanted gating effects. For Fig. 2c, the QPC gates were linearly adjusted, e.g., $\Delta V_{1u} = \Delta V_{1d} = \beta \Delta V_M$ where $\beta$ is an empirical factor keeping $g_1$ relatively constant. We first measured $g_{open}$, purple line in Fig. 2c, to confirm that the cavity is open in its transverse directions. Even when the cavity was highly open to its sides, $g_{open} \gtrsim 4$ for $V_M \gtrsim 0.15\,V$, many conductance peaks were observed from $g_{cav}$, blue line, where the quasiperiodicity strongly resembles Fabry–Pérot resonance spectra. Furthermore, a similar lineshape can be seen from $g_{loss}$ where even the finer peak structures coincide, marked with black arrows, which are likely from the transverse modes[25,34,36] to be discussed in the following figures. These qualities signify the formation of resonant modes within the open cavity, as all transmission from an optical cavity is known to be proportional to the so-called enhancement factor of the modes[26].

Prior to further analysis, we first establish a mesoscopic cavity model in the Landauer–Büttiker formalism by considering the wave amplitudes[44,45] as shown in Fig. 2d. The main conductance of interest, $g_{cav}$, is given by the squared magnitude of the cavity transmission amplitude $t_{12}$:

$$t_{12} = t_2 \frac{1}{1 - u_1 r_1 u_2 r_2} u_1 t_1 \tag{1}$$

where $t_i (r_i)$ is the transmission (reflection) amplitude of mirror $i$ and $u_i$ the amplitude acquired while propagating from mirror $i$ to the opposite mirror (see Supplementary Fig. S1 and Supplementary Note 1 for full derivation). Here, $u_i$ plays the important role of quantifying the coupling between cavity modes and the open cavity sides. In a lossless resonator cavity, the propagation is unitary and the amplitude is simply $u_i = \exp(ikL)$, where $k$ is the wavenumber, which recovers the usual

Fabry–Pérot interference. However, lossy channels, such as diffraction[25,26,46] or decoherence[47,48], leads to subunitary propagation, $|u_i|^2 < 1$, and induces a broadening in the transmission peaks. Note that this model differs from the optical framework in that the phase factor is linear not to frequency but to the wavenumber (see Discussion).

The propagation amplitude was obtained by fitting the model to $g_{cav}$ for several values of $g_{1,2}$. Henceforth, the mirror QPC conductances were kept constant by a nonlinear adjustment to the mirror gate voltages as a function of $V_M$ (see Supplementary Fig. S2). Figure 2e plots the cavity conductance for $g_{1,2} = 0.25 \sim 0.75$, and we see that the transmission power of the cavity decreases with the mirror transmission as expected. The normalized lineshape of the circled peaks have been replotted in Fig. 2f as colored dots where the solid black lines indicate the transmission coefficient $|t_{12}|^2$ for mirror transmissions $|t_{1,2}|^2 = 0.25 \sim 0.75$ predicted for a lossy cavity, $|u_{1,2}| = 0.66$. The dotted black line corresponds to a lossless cavity, $|u_{1,2}| = 1$. From comparison, it is clear that the cavity modes are strongly coupled to the open, transverse reservoirs. This quality distinguishes our open electron cavity from past reports, which required an explicit and strong transverse confinement for robust Fabry–Pérot interferences[8,9,18,49,50].

## Resonance energy spectrum

Figure 3a shows the source-drain bias spectroscopy obtained by measuring $g_{cav}$ while applying a voltage bias $V_{DC}$ to reservoir S, configuration shown in Fig. 3b. The cavity sides, O1 and O2, were floated so that the voltage drop occurred only across the mirror QPCs. From the full measurement, inset of Fig. 3a, we have analyzed the part where the conductance peaks were sharply defined with no additional fine structures. The bright regions at $V_{DC} = 0$ indicate the cavity resonances, and a finite $V_{DC}$ splits the conductance peak into negatively and positively sloped parts, each corresponding to when the modes resonate with the biased and grounded Fermi levels, respectively. Similar to the Coulomb diamond of QDs[31], the peaks trace a diamond

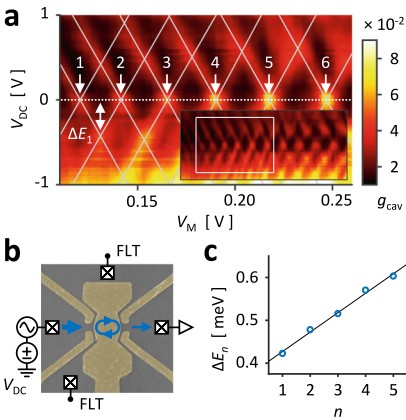

**Fig. 3 | Resonance energy spectrum. a** Source-drain bias spectroscopy of the cavity following the measurement scheme in **b**. A bias $V_{DC}$ was applied to reservoir S while keeping reservoirs O1 and O2 floated in order to restrict the voltage drop to across the mirrors. Diamond shapes (diagonal lines in **a**) are obtained by following the conductance peaks; the positively (negatively) sloped line corresponds to the cavity mode aligning with reservoir S (D). The height of the diamonds $\Delta E_n$ gives the energy spacing between the cavity modes, labeled $n$. **c** Energy spacing for peaks without additional fine structures reveal a linearity between $\Delta E_n$ and $n$, with the slope for $L = 498$ nm giving the best fit. The agreement with our sample dimension confirms that $V_M$ mainly modulates the longitudinal Fabry–Pérot interference.

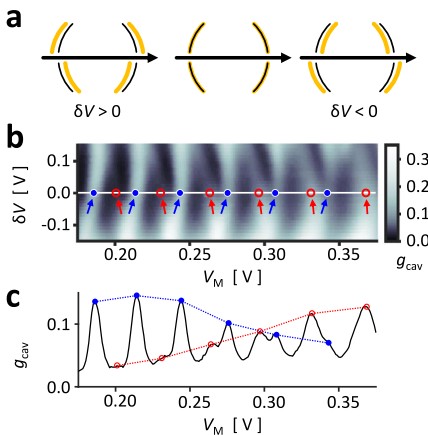

**Fig. 4 | Transverse mode detuning. a** Cavity deformation from $\delta V = V_u - V_d$ breaks the reflection symmetry across the cavity axis while maintaining the mean cavity length, introducing a major perturbation between the transverse modes while minimally affecting the longitudinal modes. **b** Transverse mode detuning observed from $g_{cav}$ measured in configuration **2a** as a function of $\delta V$, mirror conductances set to $g_{1,2} = 0.6$ at $\delta V = 0$ V. Two transverse modes were identified by their opposite evolution with respect to $\delta V$: one marked in red and the other marked in blue. **c** The dominant resonance undergoes a transition as a function of the electron wavelength.

shape (gray lines), and the height of the diamond from $V_{DC} = 0$ V gives the energy level spacing $\Delta E_n$ between the neighboring resonances, labeled as $n = 1$ and so forth. In Fig. 3c, we have plotted the level spacings and see that $\Delta E_n$ increases linearly with $n$ which implies a quadratic energy spectrum, i.e. $E_n \propto n^2$. This reflects the quadratic dispersion of free electrons: $E = (\hbar k)^2 / 2m^*$ where $m^*$ is the effective mass. In the resonator, electrons acquire a dynamic phase $2kL$ after making a roundtrip between the mirrors, and including the phase $\phi_0$ from reflections gives us the resonance condition $2k_n L + \phi_0 = 2\pi n$ and $\Delta E_n = \left( \hbar^2 \pi^2 / m^* L^2 \right) \times (n + n_0 + 1/2)$ where $n_0 = \phi_0 / \pi$. The linear fit of the data gives $L = 498$ nm which is in excellent agreement with our cavity length. That is, the quasiperiodic resonances originate from longitudinal cavity modes, i.e., Fabry–Pérot resonances.

## Transverse mode detuning

A characteristic feature of the cavity resonator is the formation of transverse modes defined exclusively by the effective confinement from the mirror curvature. These transverse modes play an especially important role in understanding cavity dynamics, such as semiclassical quantization and avoided resonance crossings[27–30,46,51–55]. We tuned the transverse modes by introducing a deformation to the mirror geometry. Consider the left mirror gates 1u-1d. In an ideal case, the cavity mirror is symmetric when the gate voltages are equal, but placing a more negative voltage on 1d pushes the lower half of the mirror into the cavity. Simultaneously placing a less negative voltage on 1u pulls the upper half of the mirror away from the cavity, maintaining the average length of the cavity as illustrated in Fig. 4a. Hence, the cavity deformation was parametrized by the voltage difference between the upper and lower mirror gates, e.g., $\delta V_1 = V_{1u} - V_{1d}$.

Figure 4b plots $g_{cav}$ for various $\delta V = \delta V_1 = \delta V_2$ where the mirror QPCs have been set to $g_{1,2} = 0.6$ at $\delta V = 0$ V. Near $\delta V \approx -0.1$ V, we see a regular set of Fabry–Pérot resonances as expected. However, the resonance peaks start splitting into two parts as we move up towards a more positive $\delta V$. One set moves leftwards (red arrows), and the other set moves rightwards (blue arrows). This continues until the two sets of peaks meet again near $\delta V \approx 0.1$ V. In optical cavities, such peak splitting is commonly understood as transverse mode detuning: each transverse mode corresponds to a set of longitudinal modes, and a perturbation to a specific transverse mode applies a phase shift to all the

corresponding Fabry–Pérot resonances. Since the cavity deformation $\delta V$ specifically breaks the reflection symmetry across the central cavity axis, we expect a detuning between modes with different transverse distributions.

Curiously, we found a wavelength-dependent transition in the dominant transverse mode. As a concrete example, the data for $\delta V = 0$ V has been replotted in Fig. 4c, where the peak positions have been marked with blue and red circles and the change in peak heights have been traced with dashed lines. As we went from lower to higher values of $V_M$, the initially dominant blue peaks diminished as the red peaks eventually dominated the cavity conductance. We expect two main factors to the transition. First, the mirror QPC can be considered a single-slit diffractor, and the wavelength modulation affects the diffraction angle of the electrons entering the cavity. Second, $\delta V$ breaks the symmetry between upper and lower gate potentials, thereby introducing an angle at which the electrons are collimated. As $V_M$ is raised, the electron wavelength decreases, and the narrower diffraction angle forces the electrons to follow the direction dictated by $\delta V$. Since transverse modes have different geometries, the dominantly excited mode would consequently undergo a transition.

## Cavity mode distribution

The transverse mode geometry was more directly probed by measuring the magnetoconductance of a detuned cavity, Fig. 5. As shown in Fig. 5b, an out-of-plane magnetic field $B$ applies a Lorentz force to the cavity electrons. For a transverse mode lying on the cavity axis, Fig. 5c, the direction of the magnetic field is chiefly irrelevant, and $g_{cav}$ responds nearly symmetrically to both positive or negative $B$. However, a lopsided transverse mode is expected to distinguish between the signs of $B$. Consider an electron launched from the left mirror, guided towards the upper half of the right mirror, as illustrated in Fig. 5d. A positive $B$ would deflect the electron upwards and increase the loss to the open side. On the other hand, a negative $B$ would guide the electron back toward the mirror center and enhance the conductance. That is, the spatial distribution of transverse modes can be known from the sign of $B$ at which $g_{cav}$ shows a maximum.

Figure 5a shows the magnetoconductance for a cavity with a positive detuning voltage $\delta V > 0$, similar to the illustrations Fig. 5c, d. As in Fig. 4, we observed two sets of Fabry–Pérot resonances.

Inspecting the data at $B = 0$ mT, we have again marked the set of peaks dominant at low $V_M$ with blue circles and those dominant at high $V_M$ with red circles. We see a clear difference in the response of the two sets; the blue-circled peaks were centered at $B \approx 0$ mT while the red-circled peaks were maximized at $B \approx -50$ mT $\sim -20$ mT (red crosses), hinting that the transverse modes of red-circled peaks were located in the upper half of the cavity. The semiclassical picture was supported by simulation as well. Figure 5e shows the simulated $g_{cav}$ of a cavity detuned to similar conditions (see Supplementary Fig. S3), calculated using the numerical tight-binding calculation package KWANT[56]. The set of conductance peaks were marked in the same manner as Fig. 5a. Then, a peak was selected from each set, marked as S for symmetric or A for asymmetric, from which the eigenchannel wavefunction from the left to the right mirrors were obtained. Figure 5f, g shows the calculated wavefunction densities with black areas indicating the mirror gate positions. In agreement with the ray trajectory analysis, the wavefunction density for peak S, Fig. 5f, lay upon the cavity axis, drawn as a black dashed line, while that of peak A, Fig. 5g, predominantly occupied the upper half of the cavity.

## Discussion

First, we emphasize how our electron cavity is distinguished from previously studied electron resonators. The electron cavity differs from a QD[31–33], as can be seen by the energy spectrum, Fig. 3. In our cavity, we estimate an electron count on the order of $n \times L^2 \approx 500$. In a QD with a similar number of electrons, the resonance energy is dominated by the electrostatic charging energy, which diminishes with each electron to a constant value because of the increased screening effect[31,33]. However, our cavity resonator is strongly coupled to its open sides, and we found no contribution from Coulomb repulsion between electrons, evident from $\Delta E_n \propto n$. Also, the Fock–Darwin spectrum of a QD assumes that the kinetic energy is comparable to the potential energy from the confinement, whereas the energy of cavity modes is purely kinetic. The presented cavity device is further distinguished from past electronic Fabry–Pérot-type resonators in that the transverse confinement was supplied solely by the effective action of the mirror curvatures. Discrete modes of a resonator require that the wave be standing in all directions, and previously reported Fabry–Pérot resonators had achieved this by forming one-dimensional channels through either the explicit use of gate confinement or transversely bound states unique to the material or phase[8,9,18,49,50]. In these cases,

however, the standing waves in the transverse direction are not defined by the cavity itself, and, therefore cannot be considered transverse modes of the open cavity. These aspects distinguish the reported cavity resonator as the advantageous platform to study electron modes in a variety of contexts, such as open systems[57–60] or non-Hermitian dynamics[29,54,61].

On the other hand, the electron cavity also differs significantly from its optical counterparts in a few key aspects. The most immediate difference is the wave dispersion. In optical cavities, the resonance spectrum is often derived in the frequency domain[26], despite the wavenumber being the origin of interferences and standing waves. This relies on the fact that photons have a linear dispersion, which is not always true for material quasiparticle. For example, the figure of merit for a cavity resonator is often given by its finesse, which quantifies the ratio between the lifetime of a mode and the time taken for the cavity photon to move back and forth the mirrors, i.e., the loss rate per roundtrip. The operational definition is instead given by the ratio of the spectral peak width and the inter-peak distance. Only for a linear dispersion is this usage unambiguous on the spectrum domain: frequency or wavenumber. Naturally, the Fabry–Pérot spectrum for a quasiparticle may read differently from that of a photon. Incidentally, the loss rate per roundtrip is immediate in the mesoscopic model, i.e., $\Pi_{i=1,2}(1 - |u_i|^2) \times (1 - |r_i|^2)$, which can easily be related to the finesse without invoking temporal concepts. Another difference lies in the particle statistics of the wavefunctions. The photon is bosonic, the electron is fermionic, and we can expect such differences to affect the dynamic properties of their respective cavities[61].

Much like its optical counterpart, the electronic cavity lends itself to numerous directions and applications. A natural extension is towards a richer implementation of electron optics, where beam splitters[4,7], lenses[1,3], and the cavity structures can be combined to reap the functionalities seen from an optical table. For example, a folded cavity with beam splitters could be used to couple multiple QDs far away from each other with higher tunneling coefficients. Another direction would be towards establishing the optical principle for matter waves, e.g. how open cavity modes are affected by their nonlinear dispersions and fermionic, or even anyonic, nature. We demonstrated the cavity at a high loss rate, ensuring the openness of the cavity, but expect that lower losses can be achieved. For reference, a well-designed cavity in a similar 2DEG sample can reach a finesse of up to < 176 limited by impurity scattering, which corresponds to a loss

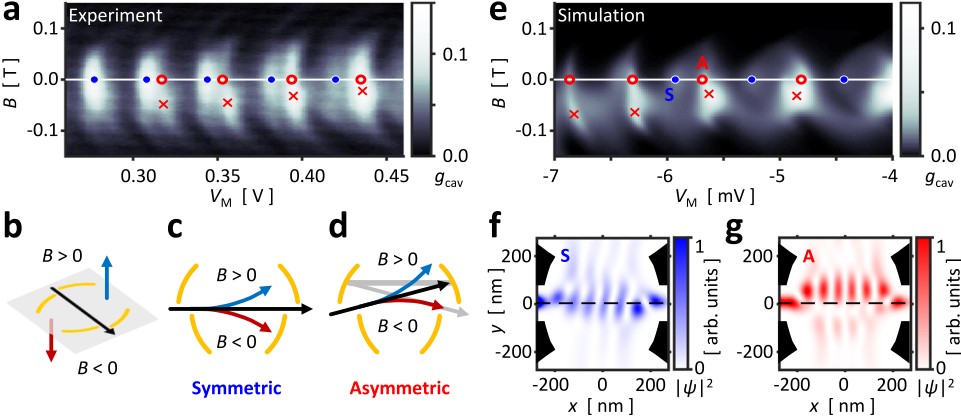

**Fig. 5 | Cavity mode distribution. a** Magnetoconductance $g_{cav}$ measured in configuration **2a** as a function of an out-of-plane magnetic field $B$ as in **b**. The field deflects the electrons either towards reservoir O1 or O2, depending on the sign of $B$. **c** For electronic modes lying on the cavity axis, a nonzero field diminishes $g_{cav}$ symmetrically for both signs of $B$ (blue circles in **a**). **d** For electronic modes lopsided across the cavity axis, $g_{cav}$ decreases for a sign of $B$ but increases for the opposite sign as the electrons are refocused towards the mirror center (red circles

in **a**). Red crosses in **a** indicate the local maxima for $g_{cav}$, where $B < 0$ indicates that the modes reside on the upper half of the cavity. **e** Numerical simulation predicts a similar magnetoconductance structure, where the conductance peaks have been marked similar to **a**. Simulated mode geometry for resonance S (A) was extracted from eigenchannel analysis and plotted in **f** (**g**), the black area and dash respectively indicating gate position and cavity axis. **f** Mode S lies on the axis. **g** Mode A is lopsided towards the upper half of the cavity.

rate of $\approx 3.5\%$ per roundtrip (see Supplementary Note 2); better figures may be achieved with a higher electron density and mobility.

In summary, we have observed the formation of electronic modes in an open cavity resonator. Despite the strong coupling to its open sides, the cavity supported well-defined modes, which were identified by resonance peaks in the conductance measurements. The regular occurrence of conductance peaks was attributed to longitudinal Fabry–Pérot modes after analyzing the conductance lineshape and resonance energy spectrum. Transverse modes were resolved by introducing a geometric deformation which induced a splitting in the conductance peaks, and the spatial distribution of the modes were investigated from the cavity magnetoconductance. With the aid of tight-binding simulations, we identified two types of transverse modes: the centered type has a conductance maximum at zero magnetic field and a wavefunction lying on the central cavity axis; the lopsided type exhibits a conductance maximum at finite magnetic fields and wavefunction predominantly occupying one side of the cavity axis. Our observations establish the electronic modes in open cavity resonators, providing a fundamental testbed for matter-wave optics.

## Methods

### Device fabrication and measurement

The experimental samples were fabricated on a GaAs/AlGaAs hetero-structure wafer, schematically shown in Fig. 1c. A layer of n-AlGaAs was grown on top of GaAs via molecular beam epitaxy[62], then the wafer was protected by a GaAs cap layer on top. In the heterostructure, dopant electrons migrate to the semiconductor interface to form a 2DEG, and the wafer used for this study had an electron density of $n = 2.3 \times 10^{11}\,\mathrm{cm}^{-2}$ with mobility $\mu = 3.6 \times 10^6\,\mathrm{cm^2/Vs}$ formed 71 nm underneath the surface. Ti/Au Schottky gates were deposited on top of the wafer via electron-beam lithography, and ohmic contacts to the 2DEG were defined via photolithography and rapid thermal annealing. Au wires leading to the gates and contacts were attached via wedge bonding.

The cavity device was measured in a homemade dilution refrigerator reaching a base temperature of $<150\,\mathrm{mK}$. All Schottky gates were biased cooled[42] with a voltage of $+300\,\mathrm{mV}$. All DC voltages on the Schottky gates and reservoirs were applied using a homemade voltage source, and conductance measurements were performed using a homemade transimpedance preamplifier followed by the lock-in amplifier SR850 (SRS). The conductance was obtained by measuring the differential current drained from device reservoirs in response to the source excitation of $10\,\mathrm{\mu V_{rms}}$ at 489 Hz. All conductances present in this report have been normalized in units of the conductance quantum $2e^2/h$ in order to emphasize the transmission properties of the device[44,45].

### Numerical simulation

The cavity device was simulated using the numerical package KWANT on Python[56]. The package simulates mesoscopic transport by numerically solving for the S-matrix of a given tight-binding model attached to semi-infinite leads. Our cavity was modeled on a spinless two-dimensional square lattice Hamiltonian $H = \sum U(ij)|ij\rangle\langle ij| - \sum t(ij;kl)|ij\rangle\langle kl|$ where $ij$ are the lattice site indices, $U$ the site-dependent onsite term, and $t$ the site-dependent hopping terms. A scattering region of size 500 nm × 500 nm was spanned using a lattice constant of $a = 5\,\mathrm{nm}$ and the hopping terms were set to $|t| = \hbar/2m^*a^2$ for nearest neighboring sites, where $m^* = 0.067 \times m_e$ is the effective mass of electrons in GaAs/AlGaAs 2DEGs for bare electron mass $m_e$. Using the package functionality, a magnetic field was applied via Peierls substitution using a gauge respecting the translation symmetry of all leads. The onsite terms were set to $U(ij) = \phi(ij) + 4|t|$ where $\phi$ is the electrostatic potential imposed by the Schottky gates positioned 50 nm above the scattering region, assuming the pinned-potential

boundary condition, and the $4|t|$ term merely repositions the 2D band minimum to zero energy. The mode wavefunctions, i.e., eigen-channels, were obtained by the singular value decomposition basis of the transmitting portion of the S-matrix[32]. All simulations in this report correspond to calculations at a Fermi energy of 8.2 meV, matching the electron density of our 2DEG. The code used in the simulation has been provided in Supplementary Software 1.

## Data availability

The raw data used in this study is provided in the supplementary information. Source data are provided with this paper.

## Code availability

The simulation code used in this study is provided in the supplementary information.

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

## Acknowledgements

We thank M. Heiblum for his experimental support and more. This work was supported by a National Research Foundation of Korea (NRF) grant funded by the Korean government (MSIP) NRF-2016R1A5A1008184 and the Quantum Computing Technology Development Program of NRF funded by the Korean government (MSIT) (No. 2019M3E4A1080144 and 2019M3E4A1080145); H.K.C. was partially supported by NRF-2022R1F1A1070970; Y.C. was partially supported by NRF-2018R1D1A1B07045946 and NRF-2020R1F1A1076284; H.J. was partially

supported by NRF-2018-Fostering Core Leaders of the Future Basic Science Program/Global Ph.D. Fellowship Program (NRF-2018H1A2A1062372); D.T.P. and H.C. were partially supported by 2019R1A2C2011538 and 2022R1A2C2010750; C.Y. was supported by NRF-2021R1I1A1A01055770.

## Author contributions

H.J. and D.T.P. contributed equally to this work. Y.C. conceptualized the study. V.U. provided the wafer material. H.J. and U.K. fabricated the device with the help of J.K. and D.K. H.J., D.T.P., U.K., and C.Y. measured the experimental data. S.L. performed the simulation. H.J., D.T.P., and S.L. analyzed the data with Y.C., H.C., H.K.C., and H.-S.S. as supervisors. H.J. and D.T.P. drafted the manuscript, which was edited by the corresponding authors.

## Competing interests

The authors declare no competing interests.
