## [Peer Review File · Nature Communications]

Observation of Electronic Modes in Open Cavity ResonatorREVIEWER COMMENTS

Reviewer #1 (Remarks to the Author):

See attached file.

The manuscript titled “Observation of Electronic Modes in Open Cavity Resonator” describes the experimental measurement and properties of 2-dimensional electron gas Fabry-Perot cavity modes in a GaAs/AlGaAs heterostructure. Cavity boundary conditions (mirrors) are constructed from metallic Schottky gates, and the wavelength of the propagating electrons can be controlled by a modulation gate, which sits over the top of the cavity. Electrons are injected into the cavity using tunnel-coupled quantum point contacts at the centre of the mirrors, and resonant modes defined by the cavity length and electron wavelength result. The mode frequencies and their losses are measured as conductances over the source and drain reservoir or between the opposite open side reservoirs, respectively. Finally, some asymmetry can be introduced into the cavity via the Schottky gate voltages which allows the resolution and identification of transverse mode polarisations, whose transmittance can be amplified or reduced via the application of a Lorentz force via an external magnetic field.

These results are an elegant demonstration of one of the fundamental properties of quantum mechanical objects; wave-particle duality, as clearly the electrons in the heterostructure are demonstrating wave-like properties such as interference and diffraction. However it is not an entirely surprising result, with very similar experimental results being presented previously [1,2,3]. The novelty of this work appears to be the claim that the cavity discussed is in fact an “open” cavity, and secondly that the transverse modes can be demarcated via the use of an external magnetic field.

In regards to the first claim, it is not immediately obvious to me what the significance of measuring an “open” cavity system is. In fact, as far as I can tell, the authors merely describe a lossy cavity, with a direct means to measure the amount of energy lost on resonance, which should relate to the resonance bandwidth. There does not seem to be anything unique about this system which would prevent its analysis using a completely classical analysis of a lossy cavity. Supposedly these results should “lay the groundwork to exploring electronic wave functions in the context of modern optical systems, such as the dielectric microcavity”, but there is no mention in the remainder of the manuscript of this microcavity or how the work presented relates to it. In my opinion, once it is accepted that the electrons in such a system indeed behave like waves and will form standing waves (which has been previously demonstrated), the results presented thereafter are trivial, or incremental at best.

For example, the second novelty of the manuscript is the ability to resolve transverse modes in the system, and through the application of a Lorentz force increase the coupling of the input and output probes to one of these modes over the other. This is an interesting result given it is not something achievable with an equivalent optical system, and the explanation given seems plausible. However, I would regard this finding as incremental. Furthermore, the manuscript fails to provide any motivation for this finding; why is it useful? On this point, I would suggest for a future version of this manuscript that an explanation be given as to why there is not a symmetric mode enhancement for $B > 0$ corresponding to a mode confined to the lower half of the resonator. Is it a consequence of the dV condition that the input coupling to such a mode is diminished i.e. in Fig 5d: why can the black line not angle downwards? Were measurements taken for $dV < 0$ which show the opposite behaviour, i.e. mode amplification of the A mode for $B > 0$? It would be useful to include this data if so.

As such, I cannot recommend this manuscript for publication in Nature Communications given the incremental nature of the research.

Some further comments to consider for any future version of the manuscript:

- A reference should be given for QPCs, it is not obvious for anyone outside of the field what they are and how they work.
- Fig. 2 (a) shows that as conductance across the gates is increased, cavity conductance also increases. The manuscript states this result is unusual given that lowering the mirror transmission i.e. (input and output coupling losses) generally leads to sharper resonances with increasingly larger transmission peaks. Firstly, it should be made more clear that lowering gate conductance is the equivalent of decreasing mirror transmission. Secondly, it appears to me that the observed phenomenon is nothing more than impedance matching the input and output ports to the cavity. The act of reducing mirror transmission should increase the cavity Q factor

(decrease cavity loss rate, κ_0) but will reduce its amplitude given that transmitted power will be proportional to $2\sqrt{\kappa_1\kappa_2}/(\kappa_0 + \kappa_1 + \kappa_2 + i\Delta\omega)$, where κ_1 and κ_2 are the loss rates introduced by the input and output ports, respectively. So while Q should increase by reducing transmission, transmitted power will not.

- Fig 2c should have its experimental data points made larger and clearer. They are very hard to see.
- Fig. 4c: why is the coupling to the two non-degenerate transverse modes wavelength dependent? Perhaps a consequence of diffraction as wavelength becomes smaller? Is there a meaning to the V_m value at which the mode dominance exchanges? It should be stated clearly on this figure that this is for the case of $\partial V=0$.

1. C G Smith *et al.*, *Fabry-Perot interferometry with electron waves*, J. Phys.: Condens. Matter **1** 9035 (1989)
2. W Liang *et al.*, *Fabry-Perot interference in a nanotube electron waveguide*, Nature **411** 665-669 (2001)
3. M T Allen *et al.*, *Observation of Electron Coherence and Fabry-Perot Standing Waves at Graphene Edge*, Nano Lett. **17**, 7380-7386 (2017)

Reviewer #2 (Remarks to the Author):

The authors reported on Electronic modes observed in an open cavity resonator. For the purpose of this study, they fabricated a simple but original device that served as a platform to demonstrate their findings.

It is clear from the very first experimental data that a gate voltage acts on the modulation of the electronic cavity and leads to the observation of multiple resonances with the proposed resonator, which is an open cavity, which seems quite new compared to previous reports dealing mostly on closed cavities.

The analogy with Fabry-Perot cavities appears as a very convincing manner to demonstrate some similarities between optical and electronic resonators. For example, the QPC conductance $g_1(2)$ are shown to be somehow equivalent to the transmission coefficients of the mirrors $t_1(2)$ and it is shown through the results shown in Fig.2 (and in line 121) how this parameter affects the finesse of the resonator (although some differences are also highlighted).

Further elaboration in the analysis is then provided with the support of a rich set of additional experimental results, as well as modeling and simulation results, along the methodology described in supplementary information, which make the whole manuscript an excellent report.

I recommend reconsidering this manuscript for publication after major revision, after addressing the following points.

I - Major comment:

There is an important clarification to be provided on the transverse modes, and under which circumstances they do appear (disappear) in the electronic resonator. Indeed, several results and statements seem contradictory:

(1st result) in the experimental data of Figure 1d, we can see that in addition to the “expected regular set of Fabry-Pérot resonances” (that is, peaks of longitudinal resonances), one can also see additional peaks of smaller amplitudes. When considering the analogy with optical Fabry-Perot cavities with curved mirrors, it is tempting to consider that those additional peaks resemble to the well-known transverse modes that can occur in such curved cavities. These transverse modes are also known as Hermite-Gaussian modes that can take place in addition to the “regular, longitudinal, Gaussian modes” (see for instance "Malak et al. Applied Physics Letters 98 (21) 2011". It is important to note that such transverse modes can occur with no need of breaking the symmetry of the mirrors, which seems similar to the condition of the results provided in Figure 1d.

(1) Please comment on these side peaks in the revised manuscript.

(2nd result) (line 151) and Fig.4 :

In Fig. 4(b) one can also see that the “rest configuration” of the cavity ($\Delta V \approx 0$ V, also corresponding to a symmetric, non-deformed cavity (as shown in Fig. 4(a)) is a configuration that leads to the co-existence of both “regular, longitudinal modes” plus “transversal modes”.

This second result seem coherent with the 1st result (and the added comments), provided that the authors make it clear in describing Figure 1d that additional peaks to the “regular set of longitudinal modes” and that those peaks probably relate to transverse modes.

At the end of this reasoning, it seems that the blue dots of Fig.4b seems to be the “regular” longitudinal modes” while the red dots are the transverse modes (contrary to what is mentioned in the caption of Figure 4(b) describing both modes as transversal modes).

(2) I disagree with this last statement that both modes are transverse modes. If needed, maybe an additional simulation could be provided for the mode shapes corresponding to those blue and red dots of Fig.4(b) (similar to those shown in Figure 5f and 5g for S and A dots, respectively) for the purpose of clarifying that the blue dots are actually longitudinal modes, not transversal modes.

(analysis of the results) After pointing-out those 2 first results, it is then extremely interesting to note that obtaining the regular set of Fabry-Pérot resonances (expression of longitudinal modes only) requires a deformation of the mirror through biasing. This fact is described by the authors in line 151) "Near $\Delta V \approx -0.1$ V, we see a regular set of Fabry-Pérot resonances as expected" (we understand that in this case, we have only longitudinal modes as shown in Figure 4(b) for $\Delta V = -0.1$ V. Such negative value of ΔV corresponds to a deformation of the cavity as illustrated in Fig.4(a).

(3) There is a lack of discussion of this result, and the quite similar situation observed at positive values of ΔV , In my opinion, both results with either negative or positive values of ΔV are due to the loss of symmetry of the mirrors along the transverse axis, leading to the gradual annihilation of the transverse modes (red dots), eventually leading to only the longitudinal modes (blue dots) to eventually continue existing. Such loss of symmetry is provided by a sufficient mirror deformation, due to a sufficiently high value of ΔV , either positive or negative.

II - Minor comments and corrections

(4) it is important to describe explicitly how the modulation of gate voltage leads to multiple resonances. It could be enough just to mention at an early stage in the manuscript that the modulation of VG, translates into a modulation of the electronic wavelength, through the action of VG on the Fermi level E_F . This clarification will certainly help the reader understanding some of the multiple aspects of the analogy with optical Fabry-Perot resonators.

(5) Many experimental details (conduction measurements, cooling, device fabrication) are missing, all those which are usually given in a "Material and Method" section. Such information can be provided as an additional supplementary section. In particular, when conducting the different electrical measurements, in order to avoid confusion, it is important to give clear information on how the different ports are connected (or left floating).

5a - For example, (line 338) for the purpose of measuring the G_{cav} conductance, please clarify, which ports have been used to probe both S and D (in this case is it between 1_u and 2_u ? between 1_d and 2_d, other ports not shown in the SEM photo of Fig. 1c ? another configuration ?

5b - Same question for g_{cav} measurements. which (metallic) ports are use to measure the conductance between O1 and O2 ?

(6) (line 90) $V_{1u}=V_{1d}=V_1+(\beta)(\Delta V_M)$ please double check this formula. In my understanding and according to Figure S2, it should be V_M instead of V_1 in the first right term of this formula.

(lines 110-111) Equation (1). The term "t" in this formula is called "t₁₂" in the supplementary information. I recommend using the same term as in the supplementary information, that is t₁₂.

Supplementary information

The open cavity resonator can be treated in the Landauer-Büttiker formalism via scattering matrices. In the supplementary information, several details seem wrong, most probably due to typo errors. Other details are missing and should be added to avoid any confusion to the reader:

(7) when introducing the succinct form Eq. S1 and defining a₁₂ and b₁₂ as matrixes involving the scalars a₁ and a₂ (b₁ and b₂), respectively, however the definitions of the matrixes A, B, r, r', and t (and t') are missing.

(8) Similarly, in Eq. S3, it is good to explicit the matrix u as a function of the scalars u₁ and u₂; and explicit the terms of the matrixes l and g. Also make it clear in the text that v is a scalar.

Expliciting the matrix form of u will also allow one to clarify the statement that $u_2 + l_2 = \dim(u)$, which seems questionable.

(9) (supplementary) line 37 "u is the transmission amplitude from one side of the cavity mirror to the other". I recommend either "transmission function" or "transmission matrix" instead of "transmission amplitude", which is misleading because u is actually not an amplitude. An alternative would be "u describes the amplitude transmission coefficients from one side of the cavity to the other".

(10) (supplementary) line 51. In the developed form of S written as a 3×3 matrix, it seems that the (multiple) denominator terms (8 terms), (all scalar) should be corrected by replacing the identity matrix I by the number 1. Same correction to be made when defining the (scalar) term V (lines 54, 55).

(11) Same comment for Eq. $S5$ and when defining $S3i$, and in many sections of the manuscript (line 56, 60, 61, etc...

(12) I suggest adding a schematic cross-sectional view of the GaAs/AlGaAs stack with some metallic patterns. Based on such new schematic and the coloured SEM Photos, it is recommended to clarify the location (use arrows when relevant) of the different building blocks of the device: in particular, (i) the space domain involving the 2D electron gas and the corresponding domain of resonant modes, (ii) the electron reservoirs, (iii) the most critical Schottky contacts and tunneling structures etc...

Reviewer #3 (Remarks to the Author):

The realization of a properties of an open resonator geometry for electrons in a GaAs/AlGaAs 2DEG and its investigation along the lines of optical micro cavities is a novel and interesting result. The author's complement their experimental measurements with relevant simulations to explain several parts of the observed mode behaviour of the cavity resonator for electron. Although several aspects of the observed mode structure remain to be understood and the results represent a significant advancement in the field.

I believe that the authors could strengthen the impact of their manuscript by addressing/commenting the following point(s):

In contrast to optical (micro) resonators, which play an very important technical role as metrological instruments (spectral filters, stable references,...) the manuscript contains little hint on where the authors see potential applications and research direction for their Fabry-Perot resonators. In this context also some comments on the limiting factors of the (scattering/diffraction dominated) cavity losses and potential projections on order of magnitude of future achievable linewidth/finesse would be helpful to underline the practical importance of the investigated system.

Further minor comments are:

Line 81: The proportionality factor α is introduced here, but no quantitative value for the system under investigation is ever provide in the manuscript, as far as I have noted.

Line 146: "Fig. 1a" should probably be "Fig. 1c"

Figure 4.: For comparison/clarity, it would be useful to state what is the value of g_1/g_2 in this detuned configuration.

In summary I believe that this work represents a significant conceptual advance with interesting insights in a topical field and merits publication in Nature Communications.

We would like to thank the reviewer for the valuable comments regarding our submission “Observation of Electronic Modes in Open Cavity Resonator.” Below, we have provided a point-by-point response in order to address the concerns or questions raised. Since the manuscript has undergone a major revision, we have quoted the first few words of the edits made in response to the comments made. A sketch of major changes has been listed on a separate document.

Reviewer #1

R1-1	These results are an elegant demonstration of one of the fundamental properties of quantum mechanical objects; wave-particle duality, as clearly the electrons in the heterostructure are demonstrating wave-like properties such as interference and diffraction. However, it is not an entirely surprising result, with very similar experimental results being presented previously [1,2,3].  1. C G Smith et al., Fabry-Perot interferometry with electron waves, J. Phys.: Condens. Matter 1 9035 (1989) 2. W Liang et al., Fabry-Perot interference in a nanotube electron waveguide, Nature 411 665-669 (2001) 3. M T Allen et al., Observation of Electron Coherence and Fabry-Perot Standing Waves at Graphene Edge, Nano Lett. 17, 7380-7386 (2017)
	The reviewer’s point about the wave-particle duality of electrons being demonstrated in the past is valid. That being said, we believe the cited literature cannot be considered analogous to the cavity resonator described in our report. A Fabry-Perot interferometer (FPI) requires a transverse confinement to collimate the waves in the longitudinal direction. The nature of said confinement determines the nature of the resonant modes, and a cavity resonator provides such confinement solely through the mirror curvature. This is why a cavity resonator can be open across the region in which free waves form resonant modes. In [1], the transverse confinement is provided by the QPC-like split gates which define a 1D channel, their main object of interest. Therefore, the resonant modes are described by a 1D wire with tunnel-coupled terminations, and the openness beyond the split gates does not affect the physics of the split gate device. In [2], the carbon nanowire also has only 1D channels. In [3], the interference occurs between graphene edge states in a magnetic field, which characterizes it as being closer to a quantum Hall edge interferometer. Since reports [1-3] use pre-defined 1D states, the optical analogy would correspond to an FPI based on optical fibers. Therefore, we consider them closer to, or fundamentally, closed systems with perturbatively tunnel-coupled leads. For example, we did not find any mention of transverse modes within the cited literature. In our device, the confinement is provided solely by the cavity mirrors, and we emphasize that the emergence of transverse modes in a truly open space is both an essential and nontrivial characteristic of the cavity resonator that hasn’t been previously demonstrated for electron waves.  • This discussion has been added to the Discussion section of the main text: “The presented cavity device is further distinguished from past electronic Fabry-Perot-type resonators...”
R1-2	The novelty of this work appears to be the claim that the cavity discussed is in fact an “open” cavity, and secondly that the transverse modes can be demarcated via the use of an external magnetic field. In regards to the first claim, it is not immediately obvious to me what the significance of measuring an “open” cavity system is. In fact, as far as I can tell, the authors merely describe a lossy cavity, with a direct means to measure the amount of energy lost on resonance, which should relate to the resonance bandwidth. There does not seem to be anything unique about this system which would prevent its analysis using a completely classical analysis of a lossy cavity. Supposedly these results should “lay the groundwork to exploring electronic wave functions in the context of modern optical

systems, such as the dielectric microcavity”, but there is no mention in the remainder of the manuscript of this microcavity or how the work presented relates to it. In my opinion, once it is accepted that the electrons in such a system indeed behave like waves and will form standing waves (which has been previously demonstrated), the results presented thereafter are trivial, or incremental at best.

We find open cavities to be significant beyond their simple lossy-ness for the following reasons. First, open cavities exhibit characteristics of non-Hermitian dynamics. For example, open cavities can exhibit nonorthogonal mode structures [4], and such properties have been instrumental to novel optical platforms, e.g. avoided resonance crossings in dielectric microcavities [5]. Second, their openness strongly affects the dynamic integrability. This distinguishes open cavities from closed ones in their manifestations of chaos [6], and also allows for the study of counterintuitive phenomena such as nonclassical diffractive billiards [7]. These avenues of research used to be unavailable, or exceedingly challenging, in electronic platforms; we believe our demonstration opens a novel and feasible direction of research using non-optical waves, specifically electron optics.

Also, the theoretical framework for optical cavities is not directly applicable to their electronic analogues. The resonance bandwidth, for example, should be the inverse lifetime of the circulating wave in optical cavities. However, the derivation relies upon the massless-ness of photons: only for linear waves does phase velocity, which determines the interference, coincide with group velocity, which determines the wave location. Therefore, it is unclear whether the usual notions used to interpret the Fabry-Perot spectrum can be applied directly to massive waves, especially quasiparticles with even more exotic band structures. We believe that our experiment using electrons provides a fundamental result towards a more general framework for cavity dynamics using nonclassical, i.e. non-optical, waves.

4. L M de Lepinay *et al.*, *Eigenmode orthogonality breaking and anomalous dynamics in multimode nano-optomechanical systems under non-reciprocal coupling*, Nature Communications **9** 1401 (1997).

5. K Jeong *et al.*, *Relative entropy as a measure of difference between Hermitian and Non-Hermitian systems*, Entropy **22** 809 (2020).

6. Sadreev, A.F., Berggren, K.F. (2006). *Signatures of quantum chaos in open chaotic billiards*. In: Khanna, F., Matrasulov, D. (eds) Non-Linear Dynamics and Fundamental Interactions. NATO Science Series II: Mathematics, Physics and Chemistry, vol **213**. Springer, Dordrecht.

7. J S Hersch *et al.*, *Influence of diffraction on the spectrum and wave functions of an open system*, Physical Review E **62** 4873 (2000)

- These references have been added to the **Discussion** section of the main text: “The presented cavity device is further distinguished from past electronic Fabry-Perot-type resonators...”

R1-3

For example, the second novelty of the manuscript is the ability to resolve transverse modes in the system, and through the application of a Lorentz force increase the coupling of the input and output probes to one of these modes over the other. This is an interesting result given it is not something achievable with an equivalent optical system, and the explanation given seems plausible. However, I would regard this finding as incremental. Furthermore, the manuscript fails to provide any motivation for this finding; why is it useful?

Since transverse modes are defined by their geometry, our magnetoconductance measurements were essential to distinguishing modes with different transverse distributions. We hope that future studies may provide more direct methods, perhaps akin to imaging techniques in optics, in order to study mode geometries in further detail.

	 This discussion has been clarified in the first sentence of the Cavity mode distribution section: “The transverse mode geometry was more directly probed...”
R1-4	On this point, I would suggest for a future version of this manuscript that an explanation be given as to why there is not a symmetric mode enhancement for $B > 0$ corresponding to a mode confined to the lower half of the resonator. Is it a consequence of the dV condition that the input coupling to such a mode is diminished i.e. in Fig 5d: why can the black line not angle downwards? Were measurements taken for $dV < 0$ which show the opposite behaviour, i.e. mode amplification of the A mode for $B > 0$? It would be useful to include this data if so.
	As seen from simulation, we found that dV induces the modes to be lopsided to only one half of the cavity. We do not have a satisfactory explanation as to why there isn't a mode corresponding to the other half of the cavity and hope that future theoretical developments may provide further insight. Unfortunately, we do not have measurements corresponding to $-dV$.
R1-5	A reference should be given for QPCs, it is not obvious for anyone outside of the field what they are and how they work. Fig. 2 (a) shows that as conductance across the gates is increased, cavity conductance also increases. The manuscript states this result is unusual given that lowering the mirror transmission i.e. (input and output coupling losses) generally leads to sharper resonances with increasingly larger transmission peaks. Firstly, it should be made more clear that lowering gate conductance is the equivalent of decreasing mirror transmission. Secondly, it appears to me that the observed phenomenon is nothing more than impedance matching the input and output ports to the cavity. The act of reducing mirror transmission should increase the cavity Q factor (decrease cavity loss rate, κ_0) but will reduce its amplitude given that transmitted power will be proportional to $2 \sqrt{\kappa_1 \kappa_2} / (\kappa_0 + \kappa_1 + \kappa_2 + i\Delta\omega)$, where κ_1 and κ_2 are the loss rates introduced by the input and output ports, respectively. So while Q should increase by reducing transmission, transmitted power will not.
	The role of QPC gate conductances has been clarified in the main text.  “The undepleted region between the $1u-1d$ ($2u-2d$) mirror gates formed a quantum point contact³⁵⁻³⁷ (QPC) with conductance g_1 (g_2), normalized in units of $G_0 = 2e^2/h$, which is equivalent to the optical transmission coefficient.” We agree that the original text contains an error regarding the transmission peak amplitude. The corresponding part of the main text has been corrected. We are thankful for the correction.
R1-6	Fig 2c should have its experimental data points made larger and clearer. They are very hard to see.
	The figure has been made clearer to see.  Fig 2f
R1-7	Fig. 4c: why is the coupling to the two non-degenerate transverse modes wavelength dependent? Perhaps a consequence of diffraction as wavelength becomes smaller? Is there a meaning to the V_m value at which the mode dominance exchanges? It should be stated clearly on this figure that this is for the case of $\partial V = 0$.
	As pointed out, we expect the wavelength dependent coupling to be a consequence of diffraction as a tunneling QPC could be considered as a single slit diffractor. A change in wavelength would affect the diffraction angle and therefore the cavity mode into which the electrons enter.

Another strong aspect seems to be that $\delta V_1 = V_{1u} - V_{1d}$ not only deforms the cavity but also redirects the direction of the QPC. However, we are cautious to provide a definite interpretation for the values of V_m and dominant modes because (1) δV only gauges the difference in the gate-induced potentials in a nominal manner and does not necessarily guarantee the (a)symmetry of the potential landscape and (2) the mode structure of electronic cavities has yet to be studied. We hope that future investigations using spatially resolved transport and theoretical understanding of electronic cavities may clarify the exact nature of the transitions observed (Fig. 4).

- This discussion has been added to the **Transverse mode detuning** section of the main text: “We expect two main factors to the transition...”
-

We would like to thank the reviewer for the detailed comments regarding our submission “Observation of Electronic Modes in Open Cavity Resonator.” Below, we have provided a point-by-point response in order to address the concerns or questions raised. Since the manuscript has undergone a major revision, we have quoted the first few words of the edits made in response to the comments made. A sketch of major changes has been listed on a separate document.

Reviewer #2

R2-1	I - Major comment: There is an important clarification to be provided on the transverse modes, and under which circumstances they do appear(disappear) in the electronic resonator. Indeed, several results and statements seem contradictory: (1st result) in the experimental data of Figure 1d, we can see that in addition to the “expected regular set of Fabry-Pérot resonances” (that is, peaks of longitudinal resonances), one can also see additional peaks of smaller amplitudes. When considering the analogy with optical Fabry-Perot cavities with curved mirrors, it is tempting to consider that those additional peaks resemble to the well-known transverse modes that can occur in such curved cavities. These transverse modes are also known as Hermite-Gaussian modes that can take place in addition to the “regular, longitudinal, Gaussian modes” (see for instance "Malak et al. Applied Physics Letters 98 (21) 2011". It is important to note that such transverse modes can occur with no need of breaking the symmetry of the mirrors, which seems similar to the condition of the results provided in Figure 1d. (1) Please comment on these side peaks in the revised manuscript.
	We agree that the additional peaks correspond to various transverse modes ubiquitous to Fabry-Perot cavities, and we also understand that our transverse modes are unlikely to be degenerate even without symmetry breaking since the cavity is not confocal [1]. Although related discussions had been relegated to later parts of the manuscript, we agree that the it would be beneficial to clarify these notes at an earlier stage. [1] G. D. Boyd and J. P. Gordon, "Confocal multimode resonator for millimeter through optical wavelength masers," in The Bell System Technical Journal, vol. 40, no. 2, pp. 489-508, March 1961, doi: 10.1002/j.1538-7305.1961.tb01626.x.  This discussion and reference have been added to the Open cavity modes section of the main text: “In a cavity resonator, multiple Fabry-Pérot spectra may be seen simultaneously when multiple transverse modes are occupied³⁵, each with their family of longitudinal modes.” and “Furthermore, a similar lineshape can be seen from g_{loss} where even the finer peak structures coincide, marked with black arrows, which are likely from the transverse modes to be discussed in following figures.”
R2-2	(2nd result) (line 151) and Fig.4 : In Fig. 4(b) one can also see that the “rest configuration” of the cavity ($\Delta V \approx 0$ V, also corresponding to a symmetric, non-deformed cavity (as shown in Fig. 4(a)) is a configuration that leads to the co-existence of both “regular, longitudinal modes” plus “transversal modes”. This second result seem coherent with the 1st result (and the added comments), provided that the authors make it clear in describing Figure 1d that additional peaks to the “regular set of longitudinal modes” and that those peaks probably relate to transverse modes. At the end of this reasoning, it seems that the blue dots of Fig.4b seems to be the “regular” longitudinal modes” while the red dots are the traverse modes (contrary to what is mentioned in the caption of Figure 4(b) describing both modes as transversal modes). (2) I disagree with this last statement that both modes are transverse modes. If needed, maybe an additional simulation could be provided for the mode shapes corresponding to those blue and red

dots of Fig.4(b) (similar to those shown in Figure 5f and 5g for S and A dots, respectively) for the purpose of clarifying that the blue dots are actually longitudinal modes, not transversal modes.

We would like to clarify our language on ‘longitudinal’ and ‘transverse’ modes as there seems to be an ambiguity in convention. Following [2], we used the terms to describe the directions in which the mode has been confined; for a cavity in two dimensions (x,y), x being the direction of the cavity axis, modes attain indices (n_x, n_y). In this notation, (n_x, 0) corresponds to the “regular, longitudinal, Gaussian modes” which are Hermite-Gaussian modes of the zeroth Hermite polynomial; and, to our understanding, the “transverse modes” of reviewer 2 seems to correspond exclusively to Hermite-Gaussian modes with higher order Hermite polynomials. We have opted to use the language of [2] in order to distinguish between the Fabry-Perot like interference, embodied by n_x, and the confinement by the mirror curvature, embodied by n_y. In this sense, all cavity modes are members of some transverse mode (n_y) each with their family of longitudinal modes (n_x), i.e. the Fabry-Perot spectra. In this convention, both blue and red modes of Fig. 4b would be transverse modes, distinguished by their responses to $\delta(V)$.

- This discussion and reference has been added to the **Open cavity modes** section of the main text: “The resonant modes of the open cavity are thus described by standing waves in two directions.”

Whether the blue or red modes correspond to the “regular longitudinal modes”, i.e. (n_x,0), we cannot say with certainty at this stage. Undoubtedly, our electrostatically defined cavity suffers from lattice impurities and realistic imperfections that are common to semiconductor fabrication, and $\delta(V) = 0V$ is only a nominal indication of cavity symmetry. In order to investigate at which $\delta(V)$ the cavity is actually symmetric, we believe one needs a method to quantify the mode geometry beyond the precision provided by our magnetoconductance measurements. As such, we believe that mode identification will indeed be a crucial next step in the study of electronic cavities.

[2] Svelto, O. (2010). Passive Optical Resonators. In: Principles of Lasers. Springer, Boston, MA. https://doi.org/10.1007/978-1-4419-1302-9_5

R2-3 (analysis of the results) After pointing-out those 2 first results, it is then extremely interesting to note that obtaining the regular set of Fabry-Pérot resonances (expression of longitudinal modes only) requires a deformation of the mirror through biasing. This fact is described by the authors in line 151 "Near $\delta(V) \approx -0.1 V$, we see a regular set of Fabry-Pérot resonances as expected" (we understand that in this case, we have only longitudinal modes as shown in Figure 4(b) for $\delta(V)=-0.1V$. Such negative value of $\delta(V)$ corresponds to a deformation of the cavity as illustrated in Fig.4(a). (3) There is a lack of discussion of this result, and the quite similar situation observed at positive values of $\delta(V)$, In my opinion, both results with either negative or positive values of $\delta(V)$ are due to the loss of symmetry of the mirrors along the transverse axis, leading to the gradual annihilation of the transverse modes (red dots), eventually leading to only the longitudinal modes (blue dots) to eventually continue existing. Such loss of symmetry is provided by a sufficient mirror deformation, due to a sufficiently high value of $\delta(V)$, either positive or negative.

As pointed out, regular Fabry-Perot resonances are seen at both positive and negative $\delta(V)$. Our naïve guess was that either the positive or negative $\delta(V)$ corresponded to a symmetric cavity, corrected for the realistic imperfections of the electronic cavity. Indeed, this would align with the opinion stated by reviewer.

However, we find this explanation somewhat mysterious. A subtle difference between our cavity deformation and optical applications is that $\delta(V)$ is pushing/pulling the upper/lower half of the mirrors. Our deformation introduces a displacement between the upper and lower mirrors on the

	order of the electron wavelength (tens to hundred nm), which is unlike the usual misalignments used in optical cavities. For example, a quarter-wavelength displacement would annihilate modes with even parity in the transverse direction (n_y is even) while those of odd parity (n_y odd) may survive. As this is a change in the total number of cavity modes, we do not know whether our deformation can be described perturbatively, i.e. whether the modes are adiabatically connected with respect to $\delta(V)$—as would be required in the reviewer’s comment. Summarizing, we do not know whether our observations can fit entirely within the conventional mode structure of optical cavities. Our belief is that a satisfactory explanation or identification of Fig. 4b requires a theoretical investigation that goes beyond the scope of a single paper. Having demonstrated the transverse mode tunability (Fig. 4) and certain spatial aspects of these modes (Fig. 5), we look forward to future investigations on the physics discussed above.
R2-4	II - Minor comments and corrections (4) it is important to describe explicitly how the modulation of gate voltage leads to multiple resonances. It could be enough just to mention at an early stage in the manuscript that the modulation of VG, translates into a modulation of the electronic wavelength, through the action of VG on the Fermi level EF. This clarification will certainly help the reader understanding some of the multiple aspects of the analogy with optical Fabry-Perot resonators.
	 The discussion has been added to the Electronic cavity resonator section of the main text: “The transmission spectra were obtained by measuring the conductances between reservoirs while modulating the wavelength of the cavity electrons by scanning through gate voltage V_M, which linearly modulates the Fermi energy^{39,40}.”
R2-5	(5) Many experimental details (conduction measurements, cooling, device fabrication) are missing, all those which are usually given in a "Material and Method" section. Such information can be provided as an additional supplementary section. In particular, when conducting the different electrical measurements, in order to avoid confusion, it is important to give clear information on how the different ports are connected (or left floating). 5a - For example, (line 338) for the purpose of measuring the Gcav conductance, please clarify, which ports have been used to probe both S and D (in this case is it between 1u and 2u? between 1d and 2d, other ports not shown in the SEM photo of Fig.1c? another configuration ? 5b - Same question for g cav measurements. which (metallic) ports are used to measure the conductance between O1 and O2?
	 We have added a Methods section in the main text explaining the experimental device The measurement schemes have been added to Figs. 2a,b and 3b.
R2-6	(6) (line 90) $V_{1u}=V_{1d}=V_1+(\beta)(\delta(V_M))$ please double check this formula. In my understanding and according to Figure S2, it should be V_M instead of V_1 in the first right term of this formula.
	We apologize for the confusion. Since the mirror transmissions are predominantly defined by V_{1u} and V_{1d}, we used $V_1 = \text{mean}(V_{1u}, V_{1d})$ as the main control for mirror 1 conductance (g_1). However, the modulation gate (V_M) affects the mirror transmission as well. So, there needs to be an appropriate correction to V_1 in order to keep the mirror transmission constant, specifically while measuring the cavity conductance against V_M. Keeping the voltages V_{1u} and V_{1d} symmetric, we hence arrive at the functional form $V_{1u}=V_{1d}=V_1+(\beta)\delta(V_M)$ where (β) is the correction factor for the changes in V_M, i.e. $\delta(V_M)$.  This has been clarified both in the main text (“$\Delta V_{1u} = \Delta V_{1d} = \beta \Delta V_M$”) and the supplementary (Fig. S2).

R2-7	(lines 110-111) Equation (1). The term “t” in this formula is called “t12” in the supplementary information. I recommend using the same term as in the supplementary information, that is t12.
	 The notation has been unified to t12 as recommended.
R2-8	Supplementary information The open cavity resonator can be treated in the Landauer-Büttiker formalism via scattering matrices. In the supplementary information, several details seem wrong, most probably due to typo errors. Other details are missing and should be added to avoid any confusion to the reader: (7) when introducing the succinct form Eq. S1 and defining a12 and b12 as matrixes involving the scalars a1 and a2 (b1 and b2), respectively, however the definitions of the matrixes A, B, r, r', and t (and t') are missing. (8) Similarly, in Eq. S3, it is good to explicit the matrix u as a function of the scalars u1 and u2; and explicit the terms of the matrixes l and g. Also make it clear in the text that v is a scalar. Expliciting the matrix form of u will also allow one to clarify the statement that $u_2 + l_2 = \dim(u)$, which seems questionable. (9) (supplementary) line 37 “u is the transmission amplitude from one side of the cavity mirror to the other”. I recommend either “transmission function” or “transmission matrix” instead of “transmission amplitude”, which is misleading because u is actually not an amplitude. An alternative would be “u describes the amplitude transmission coefficients from one side of the cavity to the other”. (10) (supplementary) line 51. In the developed form of S written as a 3x3 matrix, it seems that the (multiple) denominator terms (8 terms), (all scalar) should be corrected by replacing the identity matrix I by the number 1. Same correction to be made when defining the (scalar) term V (lines 54, 55). (11) Same comment for Eq. S5 and when defining S3i, and in many sections of the manuscript (line 56, 60, 61, etc...
	We apologize for the misunderstanding which seems to root from our implicit use of A, B, a1, a2, b1, and b2 as matrices of incoming/outgoing modes for their respective scattering components. All the variables used in the supplementary sections, e.g. r, t, u, g, and l, are matrices as well, and the explicit matrix forms naturally assumed a block form. Although the equations can be easily changed to scalars, we have left them in their current form for generality.  In order to avoid such confusion, we have explicitly mentioned that a_i and b_i are matrices in the supplementary section, expanded the term ‘amplitude’ to ‘amplitude matrix’, and added subscripts to the identity matrices to denote their dimensionalities (e.g. $\dim(I1) = \dim(A1)$). Also, we have added the following reference for the formalism used: 44. Heikkila, T. T. The Physics of Nanoelectronics: Transport and Fluctuation Phenomena at Low Temperatures. (Oxford University Press, 2013).
R2-9	(12) I suggest adding a schematic cross-sectional view of the GaAs/AlGaAs stack with some metallic patterns. Based on such new schematic and the coloured SEM Photos, it is recommended to clarify the location (use arrows when relevant) of the different building blocks of the device: in particular, (i) the space domain involving the 2D electron gas and the corresponding domain of resonant modes, (ii) the electron reservoirs, (iii) the most critical Schottky contacts and tunneling structures etc...
	 The schematic has been added to Figs 1c, e

We would like to thank the reviewer for the kind comments regarding our submission “Observation of Electronic Modes in Open Cavity Resonator.” Below, we have provided a point-by-point response in order to address the concerns or questions raised. Since the manuscript has undergone a major revision, we have quoted the first few words of the edits made in response to the comments made. A sketch of major changes has been listed on a separate document.

Reviewer #3

R3-1	In contrast to optical (micro) resonators, which play an very important technical role as metrological instruments (spectral filters, stable references, ...) the manuscript contains little hint on where the authors see potential applications and research direction for their Fabry-Perot resonators. In this context also some comments on the limiting factors of the (scattering/diffraction dominated) cavity losses and potential projections on order of magnitude of future achievable linewidth/finesse would be helpful to underline the practical importance of the investigated system.
	 The discussion has been added as the third paragraph of the Discussion section
R3-2	Line 81: The proportionality factor α is introduced here, but no quantitative value for the system under investigation is ever provide in the manuscript, as far as I have noted.
	The quantitative value of α wasn't calculated as its value was not necessary in our study, and was introduced to elucidate the role of the modulation gate.  For clarity, we have added references using a modulation gate in a similar fashion as ours: “The transmission spectra were obtained by measuring the conductances between reservoirs while modulating the wavelength of the cavity electrons by scanning through gate voltage V_M, which linearly modulates the Fermi energy^{39,40}.”
R3-3	Line 146: "Fig. 1a" should probably be "Fig. 1c"
	 The typo has been erased as the main text has undergone major revision.
R3-4	Figure 4.: For comparison/clarity, it would be useful to state what is the value of g_1/g_2 in this detuned configuration.
	 The values for g_1/g_2 have been added in both the figure caption and the main text: “Figure 4b plots g_{cav} for various $\delta V = \delta V_1 = \delta V_2$ where the mirror QPCs have been set to $g_{1,2} = 0.6$ at $\delta V = 0$”.

REVIEWERS' COMMENTS

Reviewer #1 (Remarks to the Author):

The author's have taken onboard the comments from referees and have produced a vastly improved manuscript, and clarified majority of my initial queries to a satisfactory level. Certainly, the usefulness and importance of the results has been made far more clear.

I recommend the manuscript for publication.

Reviewer #2 (Remarks to the Author):

The revised manuscript is greatly improved. Multiple points are clarified, based on the revised figures and additional supplementary information.

My opinion is that the paper can be considered for publication after a second round of minor revision, along the 2 following comments:

1 - Figure 2b. It can see a blue arrow (not a "white arrow" as written in fig.2 caption)

2 - caption of Fig2c. "The longitudinal nature of the modes can be seen by the quasi-periodicity of the conductance peaks, and hints of the transverse modes can be seen by the fine peak structures (arrows). These characteristics were observed while the cavity stayed highly open to its sides",

As pointed-out in my first report, the resonator configuration considered in this work has many similarities with the micro-structure of the (open) FP micro-cavity with curved mirrors as reported by Malak et al. that cannot totally ignored.

Although the latter work relates to an optical cavity, it has multiple similarities with the structure under consideration in this paper: it is microscale, it has curved mirrors and open architecture, it exhibits both longitudinal and transverse modes, where those transverse modes (side peaks) are also defined exclusively by the effective confinement from the mirror curvature. I still believe that it is worth

considering this work among the references at this stage of the introduction (besides the reference to G. D. Boyd and J. P. Gordon).

At this stage of the progress of the authors in their research work, I can admit that the results shown later on the manuscript, starting from Figure 3, are novel with significant scientific relevance. However, regarding the discussion on the definition of what are the transverse modes, I am not entirely convinced about the responses provided by the authors. But I understand and can admit that it is still an open question which requires further investigation.

Reviewer #3 (Remarks to the Author):

In their revised manuscript "Observation of Electronic Modes in Open Cavity Resonator", the authors have suitably addressed the points raised during my review. The rewriting, extension of the text and modifications to the figures add to making the research article more readable and instructive. Although a full characterization and understanding of the mode structure of the electronic open cavity resonators remain to be investigated in future studies, the presented results are noteworthy and of significance to the field. I thus can recommend publication of the manuscript in Nature Communications.

Reviewer #1

R1	The authors have taken onboard the comments from referees and have produced a vastly improved manuscript, and clarified majority of my initial queries to a satisfactory level. Certainly, the usefulness and importance of the results has been made far more clear. I recommend the manuscript for publication.
	We would like to thank the reviewer for the illuminating, detailed, and clarifying comments during the review process of our manuscript "Observation of Electronic Modes in Open Cavity Resonator."

Reviewer #2

R2	The revised manuscript is greatly improved. Multiple points are clarified, based on the revised figures and additional supplementary information. My opinion is that the paper can be considered for publication after a second round of minor revision, along the 2 following comments: 1 - Figure 2b. It can see a blue arrow (not a "white arrow" as written in fig.2 caption) 2 - caption of Fig2c. "The longitudinal nature of the modes can be seen by the quasi-periodicity of the conductance peaks, and hints of the transverse modes can be seen by the fine peak structures (arrows). These characteristics were observed while the cavity stayed highly open to its sides", As pointed-out in my first report, the resonator configuration considered in this work has many similarities with the micro-structure of the (open) FP micro-cavity with curved mirrors as reported by Malak et al. that cannot totally ignored. Although the latter work relates to an optical cavity, it has multiple similarities with the structure under consideration in this paper: it is microscale, it has curved mirrors and open architecture, it exhibits both longitudinal and transverse modes, where those transverse modes (side peaks) are also defined exclusively by the effective confinement from the mirror curvature. I still believe that it is worth considering this work among the references at this stage of the introduction (besides the reference to G. D. Boyd and J. P. Gordon) . At this stage of the progress of the authors in their research work, I can admit that the results shown later on the manuscript, starting from Figure 3, are novel with significant scientific relevance. However, regarding the discussion on the definition of what are the transverse modes, I am not entirely convinced about the responses provided by the authors. But I understand and can admit that it is still an open question which requires further investigation.
	1 - The typo has been corrected. 2- We see that the cavity discussed in our manuscript shares some similarities with that reported in Malak et al., albeit the latter being an optical platform. The aforementioned literature has been added to the introduction and main text under reference [34]. We also look forward to the future studies that may further clarify the nature of transverse modes in electron cavities. We would like to thank the reviewer for the illuminating, detailed, and clarifying comments during the review process of our manuscript "Observation of Electronic Modes in Open Cavity Resonator."

Reviewer #3

R3	In their revised manuscript "Observation of Electronic Modes in Open Cavity Resonator", the authors have suitably addressed the points raised during my review. The rewriting, extension of the text and modifications to the figures add to making the research article more readable and instructive. Although a full characterization and understanding of the mode structure of the electronic open cavity resonators remain to be investigated in future studies, the presented results are noteworthy and of significance to the field. I thus can recommend publication of the manuscript in Nature Communications
	We would like to thank the reviewer for the illuminating, detailed, and clarifying comments during the review process of our manuscript "Observation of Electronic Modes in Open Cavity Resonator."